# Stochastic Bandits for Egalitarian Assignment

**Eugene Lim**                                                                *elimwj@comp.nus.edu.sg*
*Department of Computer Science,*
*National University of Singapore*

**Vincent Y. F. Tan**                                                         *vtan@nus.edu.sg*
*Department of Mathematics,*
*Department of Electrical and Computer Engineering,*
*National University of Singapore*

**Harold Soh**                                                                *harold@comp.nus.edu.sg*
*Department of Computer Science,*
*National University of Singapore*

**Reviewed on OpenReview:** *https://openreview.net/forum?id=kmKVJl2JWo*

## Abstract

We study `EgalMAB`, an egalitarian assignment problem in the context of stochastic multi-armed bandits. In `EgalMAB`, an agent is tasked with assigning a set of users to arms. At each time step, the agent must assign exactly one arm to each user such that no two users are assigned to the same arm. Subsequently, each user obtains a reward drawn from the unknown reward distribution associated with its assigned arm. The agent's objective is to maximize the minimum expected cumulative reward among all users over a fixed horizon. This problem has applications in areas such as fairness in job and resource allocations, among others. We design and analyze a UCB-based policy `EgalUCB` and establish upper bounds on the cumulative regret. In complement, we establish an almost-matching policy-independent impossibility result.

## 1 Introduction

The multi-armed bandit (MAB) problem serves as a model for online decision-making under uncertainty, finding applications in diverse domains (Shen et al., 2015; Durand et al., 2018; Ding et al., 2019; Mueller et al., 2019; Forouzandeh et al., 2021). In the classical stochastic MAB problem, an agent is provided with a set of $K$ arms, each associated with an unknown distribution. At each round, the agent plays an arm and receive a reward drawn from its distribution. The agent's goal is to maximize the expected cumulative reward obtained over a fixed number of time steps $T$.

In our work, we study the problem of *egalitarian assignment* in the context of stochastic MABs, which we refer to as `EgalMAB`. In this scenario, the agent is provided with a set of $U < K$ users. At each time step, the agent must assign exactly one arm to each user such that no two users are assigned to the same arm. Subsequently, each user obtain a reward drawn from the reward distribution associated with its assigned arm. The agent's objective is to *maximize the minimum* expected cumulative reward among all users.

`EgalMAB` finds applications in various domains, including ensuring fairness in job and resource allocations. Consider the job assignment problem depicted in Figure 1. In this scenario, there are $U$ users with recurring jobs of equal load (e.g., hourly database updates) and $K$ shared cloud computing resources with fluctuating computational power (e.g., due to unrelated loads that are running on neighbouring cores of the same physical node (Kousiouris et al., 2011)). The agent's objective is to distribute these jobs *fairly* among the available resources such that over $T$ trials, the user who has to wait the longest across all their jobs (i.e., receives the least cumulative reward) is not significantly worse off compared to other users.

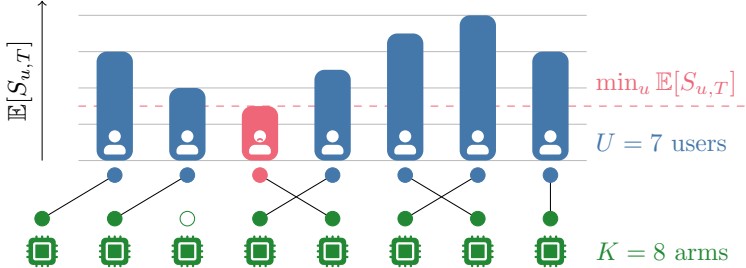

Figure 1: An illustration with $K = 8$ arms and $U = 7$ users. After time step $T$, each user $u \in [U]$ has some expected cumulative reward $\mathbb{E}[S_{u,T}]$. The agent's objective is to maximize $\min_{u \in [U]} \mathbb{E}[S_{u,T}]$, which is the minimum expected cumulative reward across all users.

Similarly, ride-hailing services present another scenario where fair assignment is desirable. In this scenario, there are $U$ passengers and $K$ drivers who offer varying degrees of experience to passengers (e.g., due to vehicle condition and driver behavior). The agent's objective is to allocate users *fairly* to vehicles such that over $T$ trips, no passenger faces a significantly worse overall experience than others.

As illustrated in both examples, `EgalMAB` embodies the principle of egalitarianism, which is also known as the Rawlsian maximin principle (Rawls, 1971), in its notion of fairness. Egalitarianism is a fundamental notion of justice based on the *difference principle*, which seeks to *maximize* the welfare of those in society who are the *worst-off*. Likewise, our agent's objective is to maximize the cumulative reward of the user receiving the lowest cumulative reward among all $U$ users. Given our examples above, solving the `EgalMAB` problem yields a policy that minimizes the overall waiting time of the user who has to wait the longest and maximizes the experience of the passenger who has the most negative encounters. This is in contrast to the classic MAB setting where only *overall utility* is considered. This prompts the fundamental question that our work seeks to answer:

> *How can we design an agent's assignment policy that optimizes overall utility for individual users while also ensuring that no user within a group consistently encounters substantially sub-optimal outcomes?*

Our contributions are summarized as follows. We formally define the `EgalMAB` problem in Section 3 and propose `EgalUCB`, a UCB-based solution to `EgalMAB`, in Section 4. In Section 5 and 6, we establish that `EgalUCB` achieves an expected regret of at most

$$O\left( \sqrt{\frac{T \ln(T) \cdot (K - U)}{U}} \right).$$

We also provide a policy-independent lower bound that matches the upper bound up to a multiplicative factor of $\sqrt{1/U}$ and a term logarithmic in $T$. Proof sketches for these results are included in Section 6. Lastly, empirical validations for these results using both synthetic and real-world data are presented in Section 7.

## 2 Related Works

**MAB with Multiple Plays.** The multiple-play multi-armed bandit (MP-MAB) problem (Anantharam et al., 1987; Gai et al., 2012; Chen et al., 2016; Komiyama et al., 2015) expands upon the classical MAB framework by allowing the agent to play $U \geq 2$ distinct arms. The MP-MAB problem has been extended in various ways, including combinatorial bandits (Cesa-Bianchi & Lugosi, 2012; Kveton et al., 2015a; Chen et al., 2016), cascading bandits (Kveton et al., 2015b; Wen et al., 2017), and MP-MAB with shareable arms (Wang et al., 2022). An adjacent problem to ours is identifying the top $U$ arms while minimizing regret, which can be framed as a matroid bandit problem (Kveton et al., 2014) with a uniform matroid of rank $U$. Although similar to `EgalMAB` in that the agent has to select multiple arms, unlike `EgalMAB`, these

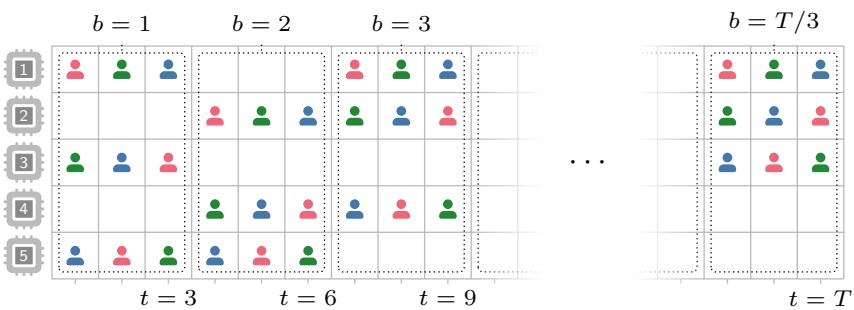

Figure 2: A trace of `EgalUCB` with $K = 5$ arms and $U = 3$ users. When $b = 2$, the three arms with the highest $\text{UCB}_{a,b}$ values are $a \in \{2, 4, 5\}$, which are then assigned in a round-robin fashion across time steps $t \in \{4, 5, 6\}$. As $b$ increases, the estimates for $\mu_a$ for all arms $a$ improves. By $b = T/3$ for large $T$, the three arms with the highest $\text{UCB}_{a,b}$ values are most likely $a \in \{1, 2, 3\}$, which are then assigned over time steps $t \in \{T - 2, T - 1, T\}$.

formulations of the MP-MAB problem do not involve *multiple users*; that is, the reward is with respect to the agent instead of users.

**MAB with Multiple Users.** The extension of multiple users into the classic MAB problem has been extensively explored in the context of cognitive radio networks (Jouini et al., 2009; Liu & Zhao, 2010; Avner & Mannor, 2014). Unlike `EgalMAB` where a *centralized* agent assigns the arms to the users, these works have focused on scenarios where multiple *decentralized* users interact with a single MAB instance. As a result of this decentralization, it is possible for multiple users to play the same arm simultaneously, resulting in a collision that can negatively affect the received reward.

**Fairness in MAB.** The introduction of fairness considerations into the classical MAB problem has garnered significant interest. Much attention has been directed towards addressing fairness concerns that typically involve ensuring that each arm is played a minimum number of times, known as fairness in exposure (Claure et al., 2020; Chen et al., 2020; Li et al., 2020; Wang et al., 2021), or with a probability proportional to the arm's merit, known as meritocratic fairness (Joseph et al., 2016; 2018). While `EgalMAB` emphasizes fairness among users, the aforementioned works deal with fairness among arms. However, there are alternative approaches that maintain a focus on fairness among users. One such approach involves maximizing the Nash social welfare (NSW) function. This function is defined as the product of rewards obtained by all users, which intuitively encodes the notion of fairness as it increases only when most users achieve high rewards. Hossain et al. (2021) investigated a scenario involving $U$ users and $K$ arms, in which each arm's reward varies for each user due to different perceived utilities. In each time step, only one arm is played, and all users obtain rewards according to their perceived utilities. Their primary objective is to maximize the NSW. Sawarni et al. (2023) examined an alternative fairness framework within the context of linear bandits. In their scenario, a new user is introduced at each time step, and fairness is ensured by considering the product of rewards across all time steps. Both of these works assume that only a *single* arm is played at each time step, while the agent in `EgalMAB` simultaneously selects *multiple* arms. Additionally, instead of aiming to maximize the *NSW function*, `EgalMAB` focuses on maximizing the *cumulative reward of the worst-off user*.

## 3  `EgalMAB` **Problem**

In this section, we will formally define the components involved in an `EgalMAB` problem. We will start by presenting a working definition for each component. When relevant, we will supplement these definitions with the measure-theoretic details necessary for the proofs of lower bounds.

**Environment.** Let $K$ be the number of arms, $U < K$ be the number of users, and $T$ be the time horizon. For each arm $a \in [K]$, let $p_a$ denote the reward density. An instance of the `EgalMAB` problem is represented by the tuple $(\nu, U, T)$ where $\nu := (p_1, \ldots, p_K)$. When the context is clear, we also refer to $\nu$ as an `EgalMAB` instance. We assume that the expected reward $\mu_a := \mathbb{E}_{X \sim p_a}[X]$ obtained for playing arm $a$ is finite. Moreover, for convenience, we assume that the arms are indexed in such a way that $\mu_1 \geq \cdots \geq \mu_K$. However, the agent is unaware of this ordering.

More formally, for each arm $a \in [K]$, let $\mathbb{P}_a$ be the probability law for the reward obtained after playing arm $a$. For any $\mathbb{P}_a$-measurable set $B$, we have $\mathbb{P}_a(B) = \int_B p_a \, d\lambda$, where $\lambda$ is the Lebesgue measure and $p_a = d\mathbb{P}_a/d\lambda$ is the Radon–Nikodym derivative for $\mathbb{P}_a$.

**Agent Policy.** A solution to the `EgalMAB` problem is characterized by an agent policy $\pi := (\pi_t)_t$. During each time step $t \in [T]$, the map $\pi_t$ considers the actions and rewards history (which we will define shortly, after introducing relevant notations) and assigns each user $u \in [U]$ to an arm $A_{u,t} \in [K]$ such that no two users are assigned the same arm. Subsequently, each user $u$ receives a reward $X_{u,t}$ drawn independently from the density $p_{A_{u,t}}$. Using these notations, we will denote the history that $\pi_t$ considers as $(A_1, X_1, \ldots, A_{t-1}, X_{t-1})$ where $A_t := (A_{u,t})_u$ and $X_t := (X_{u,t})_u$.

More formally, let $\mathcal{P}(S)$ denote the power set of a set $S$. For each time step $t \in [T]$, the stochastic map

$$\pi_t : (([K] \times \mathbb{R})^U)^{t-1} \times \mathcal{P}([K]^U) \to [0,1]$$

is a probability kernel. We denote $\mathbb{P}_{\pi\nu}$ as the probability law for the interaction between the policy $\pi$ and the `EgalMAB` instance $\nu$ over $T$ time steps. Thus, the density of $\mathbb{P}_{\pi\nu}$ is

$$p_{\pi\nu}(A_1, X_1, \ldots, A_T, X_T) = \prod_{t=1}^{T} \pi_t(A_t | A_1, X_1, \ldots, A_{t-1}, X_{t-1}) \prod_{u=1}^{U} p_{A_{u,t}}(X_{u,t}).$$

**Egalitarian Objective.** To achieve egalitarian fairness, we want to design a policy $\pi$ that maximizes the expected cumulative reward of the least-rewarded user. Let

$$S_{u,t} := \sum_{s=1}^{t} X_{u,s}$$

be the cumulative reward for user $u$ up to time $t$. Formally, our egalitarian objective is to maximize $\min_{u \in [U]} \mathbb{E}[S_{u,T}]$. However, in line with most works in MAB, we will frame our problem as regret minimization. We define the *expected cumulative regret* as

$$R_T := \frac{T\mu_*}{U} - \min_{u \in [U]} \mathbb{E}[S_{u,T}]$$

where $\mu_* := \mu_1 + \cdots + \mu_U$ is the sum of the expected reward of the top $U$ arms. The choice to compare with $T\mu_*/U$ is natural because the maximum expected cumulative reward obtained by the user with the least reward is at most $T\mu_*/U$, which is obtained by pulling the best arms in a round robin. To see this, observe that the sum of cumulative rewards for all $U$ users is $\sum_u \mathbb{E}[S_{u,T}] \leq T\mu_*$. Suppose, to the contrary, that $\min_u \mathbb{E}[S_{u,T}] > T\mu_*/U$, then $\sum_u \mathbb{E}[S_{u,T}] > T\mu_*$, which is a contradiction.

# 4 `EgalUCB` Policy

In this section, we describe our policy `EgalUCB` that achieves near-optimal regret for the `EgalMAB` problem. The `EgalUCB` policy is based on the `UCB1` policy (Auer et al., 2002) for solving classic MAB problems; however, it differs from the `UCB1` in several key aspects. We present the pseudocode for `EgalUCB` in Algorithm 1, complemented by a visual guide in Figure 2. We also provide an alternate version of the pseudocode with more implementation details in Appendix B.

---

**Algorithm 1:** `EgalUCB`

---

**1** let number of blocks $B_{a,0} = 0$, cumulative reward $S_{a,0} = 0$, and $\text{UCB}_{a,0} = \infty$ for each $a \in [K]$

**2** **for** $b = 1, 2, \dots, T/U$ **do**

**3** $\quad$ let $A_b \subseteq_U [K]$ be a set of $U$ arms with highest $\text{UCB}_{a,b-1}$

**4** $\quad$ play arms $A_b$ in a round robin fashion for the next $U$ steps

**5** $\quad$ update $B_{a,b}$ and $S_{a,bU}$ accordingly for each $a \in A_b$

**6** $\quad$ update $\text{UCB}_{a,b} = \dfrac{S_{a,bU}}{B_{a,b}U} + \sqrt{\dfrac{6\ln(bU)}{B_{a,b}U}}$ for each $a \in A_b$

**7** **end**

---

`EgalUCB` partitions the horizon into $B := T/U$ blocks, each with $U$ steps. We assume, without loss of generality, that $T$ is divisible by $U$. In cases where this is not true, the difference in expected regret between the best and worst user is at most $(\mu_1 + \cdots + \mu_{\lceil U/2 \rceil}) - (\mu_{K-\lfloor U/2 \rfloor} + \cdots + \mu_U)$, which is independent of $T$.

Let $A \subseteq_U S$ denote that set $A$ is a subset of set $S$ with size $U$. `EgalUCB` begins by initializing some statistics (Line 1). At the start of each block $b \in [B]$, it selects any set $A_b \subseteq_U [K]$ consisting of the highest-ranked $U$ distinct arms as determined by their upper confidence bounds $\text{UCB}_{a,b}$ (Line 3). Over a block with $U$ steps, these arms are then assigned to the $U$ users in a round-robin fashion (Line 4). After observing the rewards, `EgalUCB` updates its statistics (Lines 5–6).

Since each user is assigned to every arm in $A_b$ exactly once, we have $\mathbb{E}[S_{u,T}] = \mathbb{E}[S_{u',T}]$ for all $u, u' \in [U]$. From a technical standpoint, this simplifies the regret by eliminating the min operator in the regret $R_T$:

$$R_T = \frac{T\mu_*}{U} - \min_{u \in [U]} \mathbb{E}[S_{u,T}] = \frac{T\mu_*}{U} - \mathbb{E}[S_{u,T}], \quad \forall u \in [U].$$

## 5 Main Results

In this section, we present our main theoretical results. We first state some necessary definitions and notations. Then, we present the regret upper bounds for the `EgalUCB` policy. Following that, we discuss the policy-independent regret lower bound for `EgalMAB`.

Let $X_{a,t}$ be the reward obtained from playing arm $a$ for the $t$-th time across all users. Note that if user $u$ plays arm $a$ at time step $t$, then the reward obtained is $X_{u,t} = X_{a,T_{a,t}}$ where $T_{a,t}$ is the number of times arm $a$ is played up till time step $t$.

Let $B_{a,b}$ be the number of blocks that arm $a$ is played up till block $b$. Note that $B_{a,b} = T_{a,bU}$. Furthermore, let

$$\hat{\mu}_{a,b} := \frac{1}{bU} \sum_{t=1}^{bU} X_{a,t}$$

be the empirical estimate of $\mu_a$ after playing arm $a$ for $b$ blocks. As the policy observe more rewards, it gains confidence about its estimate of $\mu_a$. This level of confidence is captured by

$$\epsilon_{b,b'} := \sqrt{\frac{6\ln(bU)}{b'U}},$$

which is the confidence radius of playing an arm for $b'$ blocks after block $b$.

Denote $A_* := [U] \subseteq_U [K]$ as the set of best $U$ arms. For any set of arms $A \subseteq [K]$ of not necessarily size $U$, let $A^- := A \backslash A_*$ be the set $A$ with the best $U$ arms removed. Furthermore, let the expected reward of $A$ be

$$\mu_A := \sum_{a \in A} \mu_a.$$

The sub-optimality gap of $A$ is defined as $\Delta_A \coloneqq \mu_{A_*} - \mu_A$. In the extreme cases, the maximum sub-optimality gap

$$\Delta_{\max} \coloneqq (\mu_1 + \cdots + \mu_U) - (\mu_{K-U+1} + \cdots + \mu_K)$$

is obtained by selecting the worst set of $U$ arms, and the minimum non-zero sub-optimality gap $\Delta_{\min} \coloneqq \mu_U - \mu_{U+1}$ is obtained by replacing arm $U$ in $A_*$ with arm $U + 1$, assuming that $\mu_U \neq \mu_{U+1}$.

Let $\nu = (p_1, \ldots, p_K)$ be an instance of `EgalMAB`. We say that $\nu$ is a 1-subgaussian `EgalMAB` if for all arms $a \in [K]$, $X \sim p_a$ is a 1-subgaussian random variable. Theorems 1 and 2 respectively provide problem-dependent and problem-independent upper bounds for the expected cumulative regret of running `EgalUCB` on a 1-subgaussian `EgalMAB`.

**Theorem 1** (Problem-Dependent Upper Bound). *Let $(\nu, T, U)$ a 1-subgaussian `EgalMAB`. After running `EgalUCB` for $T$ time steps, we have*

$$R_T \leq \frac{2136(K - U)\ln(T)}{\Delta_{\min}} + \frac{4K\Delta_{\max}}{U}.$$

**Theorem 2** (Problem-Independent Upper Bound). *Let $(\nu, T, U)$ be a 1-subgaussian `EgalMAB` with $\mu_a \in [0, 1]$ for all arms $a \in [K]$. After running `EgalUCB` for $T$ time steps, we have*

$$R_T \leq \sqrt{\frac{8544(K - U)\, T \ln(T)}{U}} + \frac{4K\min\{U, K - U\}}{U}.$$

When the number of arms $K$ and users $U$ are fixed, the problem-independent upper bound increases with the number of time steps $T$ at a rate of $O(\sqrt{T\ln(T)})$. Notably, when $U = 1$, the `EgalMAB` instance and the `EgalUCB` policy reduce to the classic MAB instance and the `UCB1` policy (Auer et al., 2002). Consequently, both the problem-dependent and problem-independent upper bounds can be reduced to the bound for classic `UCB1` with minimal effort[1].

Next, we examine how the number of users $U$ affects performance. Consider some fixed time horizon $T$ and number of arms $K$. Since $T \gg K$, the first terms in both the problem-independent and problem-dependent upper bounds dominates the regrets. Furthermore, when $K = U$, every user would have played every arm exactly once after each block, yielding an expected cumulative reward of $b\mu_*$ after block $b$. By definition, this implies that the expected cumulative regret $R_T = 0$. This behavior is reflected in both upper bounds, since $K - U = 0$ and $\Delta_{\max} = 0$ when $K = U$.

Additionally, the problem-independent upper bound decreases as $U$ approaches $K$, and this reduction scales with $O(1/\sqrt{U})$. There are two reasons for this. Firstly, if we fix some `EgalMAB` instance $\nu$ and vary $U$, then increasing $U$ results in decreasing $T\mu_*/U$. Secondly, since `EgalUCB` assigns the arms in a round-robin fashion during each block, as long as the UCB values of the top arms are consistently among the highest regardless of their order, `EgalUCB` will also consistently select a good set of arms. This implies that as $U$ increases, this problem becomes more statistically robust to the variability inherent in the estimates of the arms. Loosely speaking, the more users we have, the easier it is to match the performance of a policy always plays round-robin the set of arms $A_*$.

We further consider the scope for algorithmic improvement by deriving a policy-independent lower bound. Let $\nu = (p_1, \ldots, p_K)$ be an `EgalMAB` instance and $\pi$ be any policy. We denote $R_{\pi\nu}$ as the expected cumulative regret of running $\pi$ on $\nu$ for $T$ time steps. Theorem 3 provides a policy-independent lower bound for the regret $R_{\pi\nu}$. This bound applies to the class $\mathcal{V}$ of all Gaussian `EgalMAB` instances $\nu = (p_1, \ldots, p_K)$ where, for all $a \in [K]$, the reward density $p_a = \mathcal{N}(\mu_a, 1)$ and $\mu_a \in [0, 1]$ .

**Theorem 3** (Policy-Independent Lower Bound). *Suppose $K \geq 2U$. For any policy $\pi$, there exist an `EgalMAB` instance $\nu \in \mathcal{V}$ with regret*

$$R_{\pi\nu} \geq \frac{\sqrt{(K - U)\, T}}{76U}.$$

---

[1]Refer to the notes at the end of Lemma 5 for the details.

The lower bound suggests that `EgalUCB` is tight in $T$ up to logarithmic factors. This factor is expected when using a UCB-based policy due to the choice of confidence radius $\epsilon_{b,b'}$. Drawing parallels to the MOSS policy (Audibert & Bubeck, 2009) in classic $K$-armed MAB problems, we also conjecture that there exists a MOSS-based policy that can shave away the $\sqrt{\log T}$ term in the regret bound.

Furthermore, there is a multiplicative gap of $1/\sqrt{U}$ between the lower bound and the problem-independent upper bound. Our experimental results in Section 7 suggest that our upper bound analysis is not tight, as we empirically observe the $O(1/U)$ behavior when running `EgalUCB`.

## 6 Regret Analysis

In this section, we provide a proof sketch for the main results in Section 5. Our analysis relies on techniques developed in the combinatorial semi-bandits literature (Kveton et al., 2015a). Detailed proofs for the upper bounds and lower bound can be found in Appendices C and D respectively.

### 6.1 Problem-Dependent Upper Bound

The regret upper bound in Theorem 1 consists of a sum of two terms: the first term is dependent on $T$ and the second is independent of $T$. The first (resp. second) term arises from the regret accumulated over blocks where some *good* event $\mathcal{E}_b$ occurs (resp. did not occur). This *good* event $\mathcal{E}_b$ is the event that $\mu_a$ and its estimate $\hat{\mu}_a$ are at a distance of at most $\epsilon_{b-1,B_{a,b-1}}$ at the beginning of block $b$ for all $a \in [K]$. Formally, we define

$$\mathcal{E}_b := \bigcap_{a=1}^{K} \left\{ \left| \hat{\mu}_{a,B_{a,b-1}} - \mu_a \right| \leq \epsilon_{b-1,B_{a,b-1}} \right\}.$$

Note that we will often abuse the set notation when defining events. In particular, when we have a proposition $P$, we use $\mathcal{E} = \{P\}$ to signify that $\mathcal{E}$ is the set of all outcomes in the underlying probability space where $P$ holds. Lemma 1 shows that $\mathcal{E}_b$ occurs with high probability.

**Lemma 1.** *Let $(\nu, T, U)$ be a 1-subgaussian `EgalMAB`. Then, for all blocks $b \in [B]$,*

$$\mathbb{P}(\mathcal{E}_b^c) \leq \frac{2K}{b^2 U^3}.$$

To facilitate analysis, we then consider another *good* event $\mathcal{F}_b$ for which the set of arms $A_b \subseteq_U [K]$ selected during block $b$ is sub-optimal but "not too bad". This is defined as

$$\mathcal{F}_b := \left\{ 0 < \Delta_{A_b} \leq 2 \sum_{a \in A_b^-} \epsilon_{B,B_{a,b-1}} \right\}.$$

Lemma 2 shows that if the set of arms $A_b$ played during block $b$ is sub-optimal and $\mathcal{E}_b$ occurs, then $\mathcal{F}_b$ must follow.

**Lemma 2.** *Let $b \in [B]$. If the set of arms $A_b$ played during block $b$ is sub-optimal and $\mathcal{E}_b$ occurs, then $\mathcal{F}_b$ also occurs.*

We proceed by partitioning the regret into two terms: one conditioned on the high-probability event $\mathcal{F}_b$ using Lemma 2 and another conditioned on the low-probability event $\mathcal{E}_b^c$ using Lemma 1. This result is formally stated in Lemma 3.

**Lemma 3.** *Let $(\nu, T, U)$ be a 1-subgaussian `EgalMAB`. Then, after $T$ time steps, for all users $u \in [U]$, we have*

$$R_{u,T} \leq \sum_{b=1}^{B} \mathbb{E}[\Delta_{A_b} \mathbb{I}\{\mathcal{F}_b\}] + \frac{\pi^2 K \Delta_{\max}}{3 U^3}.$$

Let us now focus on bounding the contributions made by the high-probability term (i.e., the first term above). We first partition $\mathcal{F}_b$ into countably many mutually exclusive events $\{\mathcal{G}_{b,i}\}_i$ so that we can write

$$\Delta_{A_b}\mathbb{I}\{\mathcal{F}_b\} = \sum_{i=1}^{\infty} \Delta_{A_b}\mathbb{I}\{\mathcal{G}_{b,i}\}, \qquad \text{almost surely.}$$

To define the events $\{\mathcal{G}_{b,i}\}_i$, let $\alpha = 0.13$, $\beta = 0.22$, and

$$\gamma = 24\left(\frac{1-\beta}{\sqrt{\alpha}-\beta}\right)^2$$

be constants that are carefully chosen, and define

$$m_{b,i} := \frac{\gamma\alpha^i U \ln(BU)}{\Delta_{A_b}^2}$$

for all $i \in \mathbb{N}$. Let

$$L_{b,i} := \{a \in A_b^- : B_{a,b-1} \leq m_{b,i}\}$$

to be the set of arms in $A_b^-$ that are played for fewer than $m_{b,i}$ blocks at the beginning of block $b$. For convenience, let $L_{b,0} = A_b^-$. For each $b \in [B]$ and $i \in \mathbb{N}$, the event $\mathcal{G}_{b,i}$ is then defined as

$$\mathcal{G}_{b,i} := \bigcap_{j=1}^{i-1}\{|L_{b,j}| < \beta^j U\} \cap \{|L_{b,i}| \geq \beta^i U\} \cap \{\Delta_{A_b} > 0\}.$$

It is clear that at most one of $\{\mathcal{G}_{b,i}\}_i$ can occur. However, to show that it is a partition for $\mathcal{F}_b$, we also need to show that at least one of $\{\mathcal{G}_{b,i}\}_i$ must happen. This is shown in Lemma 4.

**Lemma 4.** *Assume that $b > K/U$. On the event $\mathcal{F}_b$, exactly one of the events in $\{\mathcal{G}_{b,i}\}_i$ occurs.*

Using the newly-defined events $\{\mathcal{G}_{b,i}\}_{b,i}$, we bound the contributions of the the high-probability term in Lemma 5. The intuition behind the events $\{\mathcal{G}_{b,i}\}_{b,i}$, which is a common construction used in the proof of combinatorial semi-bandits Kveton et al. (2015a), is that it serves to upper bound the high probability term in the regret by the number of times the set of arms $A_b$ is played. This allows us to introduce the reciprocal of the gap term for individual arms $\Delta_{a,N_a}$ which then serves to derive a meaningful problem-dependent bound on the regret.

**Lemma 5.** *Let $\nu = (p_1, \ldots, p_K)$. Suppose that $p_a$ is the density for a 1-subgaussian distribution for all $a \in [K]$. Then, after $T$ time steps,*

$$\sum_{b=1}^{B}\mathbb{E}[\Delta_{A_b}\mathbb{I}\{\mathcal{F}_b\}] = \sum_{i=1}^{\infty}\sum_{b=b_0}^{B}\Delta_{A_b}\mathbb{I}\{\mathcal{G}_{b,i}\} + \sum_{b=1}^{b_0-1}\Delta_{A_b}\mathbb{I}\{\mathcal{F}_b\}$$

$$\leq 2136\ln(BU)\sum_{a\in\Lambda}\frac{1}{\Delta_{a,N_a}} + \frac{K\Delta_{\max}}{U}$$

$$\leq \frac{2136(K-U)\ln(BU)}{\Delta_{\min}} + \frac{K\Delta_{\max}}{U}.$$

*for all users $u \in [U]$.*

The proof of Theorem 1 simply involves substituting the result of 5 into Lemma 3.

*Proof of Theorem 1.* We can bound the regret by

$$R_{u,T} \leq \sum_{b=1}^{B}\mathbb{E}[\Delta_{A_b}\mathbb{I}\{\mathcal{F}_b\}] + \frac{\pi^2 K\Delta_{\max}}{3U^3} \leq \frac{2136(K-U)\ln(BU)}{\Delta_{\min}} + \frac{K\Delta_{\max}}{U} + \frac{\pi^2 K\Delta_{\max}}{3U^3}$$

$$\leq \frac{2136(K-U)\ln(T)}{\Delta_{\min}} + \frac{4K\Delta_{\max}}{U}.$$

where the first inequality holds due to Lemma 3 and the second inequality holds due to Lemma 5. $\qquad\square$

## 6.2 Problem-Independent Upper Bound

Much like the expression in Theorem 1, the first term of the regret bound in Theorem 2 arises from the cumulative regret accumulated in blocks where the high-probability event $\mathcal{E}_b$ occurs. Conversely, the second term arises from the regret accumulated in the initial blocks and the blocks where $\mathcal{E}_b$ did not occur. The proof of Theorem 2 can be found in Appendix C.

## 6.3 Policy-Independent Lower Bound

Our proof for Theorem 3 consists of constructing two `EgalMAB` instances $\nu$ and $\nu'$ that are "close" enough that it is difficult for any policy to distinguish between them statistically, yet "far" enough that a sequence of actions that is good for one instance is bad for the other. Specifically, let

$$\Delta = \sqrt{\frac{K-U}{8TU^2}}$$

and set $\nu = (p_1, \ldots, p_K) \in \mathcal{V}$ to be an `EgalMAB` instance with

$$\mu_a = \begin{cases} \Delta, & \text{if } a \in [U] \\ 0, & \text{otherwise.} \end{cases} \tag{1}$$

Assuming that $K \geq 2U$, let

$$A' = \operatorname*{arg\,min}_{A \subseteq_U [K] \setminus [U]} \sum_{a \in A} \mathbb{E}_{\pi\nu}[T_{a,T}]$$

be the set of $U$ arms that are sub-optimal under $\nu$ have been played the fewest number of times under the distribution $\mathbb{P}_{\pi\nu}$. Set $\nu' \in \mathcal{V}$ to be an `EgalMAB` instance with

$$\mu'_a = \begin{cases} 2\Delta, & \text{if } a \in A' \\ \mu_a, & \text{otherwise.} \end{cases} \tag{2}$$

Let $\boldsymbol{\Gamma}$ be the set of size-$U$ subsets of $[K]$ that contains at least $U/2$ arms from $[U]$, and let

$$\mathcal{H} := \left\{ \sum_{A \in \boldsymbol{\Gamma}} T_{A,T} \leq \frac{T}{2} \right\}$$

be the event that, for at least $T/2$ time steps, the policy $\pi$ selects a set in which at least half of it consists of arms from $[U]$. Lemma 6 uses the Bretagnolle–Huber inequality with $\mathcal{H}$ to reduce the lower bound computation to evaluating the KL-divergence between $\mathbb{P}_{\pi\nu}$ and $\mathbb{P}_{\pi\nu'}$.

**Lemma 6.** *Let $\nu$ and $\nu'$ be the `EgalMAB` instances defined by equation 1 and equation 2. Under the assumptions of Theorem 3, we have*

$$R_{\pi\nu} + R_{\pi\nu'} > \frac{\Delta T}{4} \left( \mathbb{P}_{\pi\nu}(\mathcal{H}) + \mathbb{P}_{\pi\nu'}(\mathcal{H}^c) \right)$$
$$\geq \frac{\Delta T}{8} \exp\left( -D_{\mathrm{KL}}(\mathbb{P}_{\pi\nu} \| \mathbb{P}_{\pi\nu'}) \right).$$

Part of the novelty in our analysis involves reducing the computation of this KL-divergence to a combinatorial problem. Lemma 7 decompose the KL-divergence from Lemma 6 into an expression that involves counting the number of times each arm $a \in A'$ is played, and Lemma 8 shows, using a counting argument, that the number of times an arm $a \in A'$ is played is at most $TU^2/(K-U)$.

**Lemma 7.** *Let $\nu$ and $\nu'$ be the `EgalMAB` instances defined by equation 1 and equation 2. Under the assumptions of Theorem 3, we have*

$$D_{\mathrm{KL}}(\mathbb{P}_{\pi\nu} \| \mathbb{P}_{\pi\nu'}) \leq 4\Delta^2 \sum_{a' \in A'} \mathbb{E}_{\pi\nu}[T_{a',T}].$$

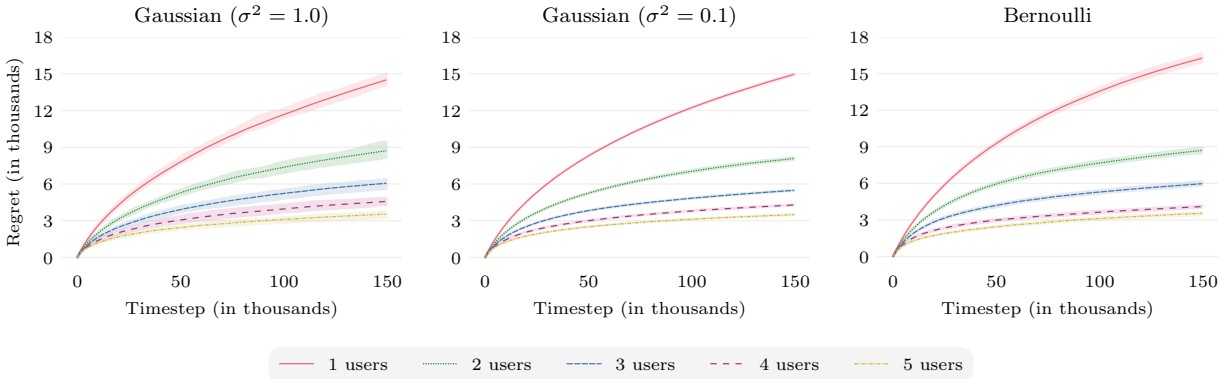

Figure 3: Expected regret incurred by `EgalUCB` over $T = 150{,}000$ time steps on simulated data with $K = 10$. Each line corresponds to a different $U$. The lighter region around each line represents the range between the minimum and maximum expected regrets observed over a total of 30 independent runs.

**Lemma 8.** *Let $\nu$ and $\nu'$ be the `EgalMAB` instances defined in equation 1 and equation 2. Under the assumptions of Theorem 3, we have*

$$\sum_{a \in A'} T_{a,T} \leq \frac{TU^2}{K - U}$$

*almost surely.*

The proof of Theorem 3 simply involves substituting the results of Lemma 7 and Lemma 8 into Lemma 6.

*Proof of Theorem 3.* We have

$$R_{T,\pi,\nu} + R_{T,\pi,\nu'} \geq \frac{\Delta T}{8} \exp\left(-4\Delta^2 \sum_{a' \in A'} \mathbb{E}_{\pi\nu}[T_{a',T}]\right) \geq \frac{\Delta T}{8} \exp\left(-\frac{4\Delta^2 TU^2}{K - U}\right)$$

$$= \frac{\Delta T}{8} \exp(-1/2) > \frac{\sqrt{T(K - U)}}{38U}.$$

Since $2 \max\{R_{T,\pi,\nu}, R_{T,\pi,\nu'}\} \geq R_{T,\pi,\nu} + R_{T,\pi,\nu'}$, dividing by 2 concludes the proof. $\square$

## 7 Experiments

In this section, we present the results of our numerical experiments to validate the analysis of `EgalUCB`. These experiments include both a synthetic environment (Section 7.1) and real-world datasets (Sections 7.2 and 7.3). The code for these experiments can be found in the supplementary materials.

### 7.1 Synthetic Experiments

To empirically verify the problem-independent upper bound in Theorem 2, we conducted experiments on three synthetic datasets: Gaussian bandits with variance $\sigma^2 = 1.0$, Gaussian bandits with variance $\sigma^2 = 0.1$ and Bernoulli bandits.

In each environment, we fixed $K = 10$ and varied $U$ from 1 to 5. For each choice of $U$, we ran the experiment 30 times. In each run, we randomly generated the ground truth expected reward $\mu_a$ for each arm $a$ using a uniform distribution with support $[0.01, 0.99]$. Figure 3 shows the expected regret incurred by `EgalUCB` over $T = 150{,}000$ time steps. As predicted by Theorem 2, the expected regret $R_T$ is sub-linear in $T$ and diminishes with $U$.

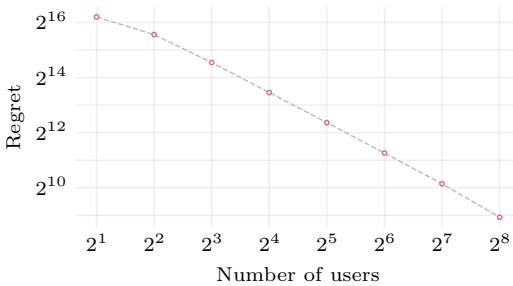

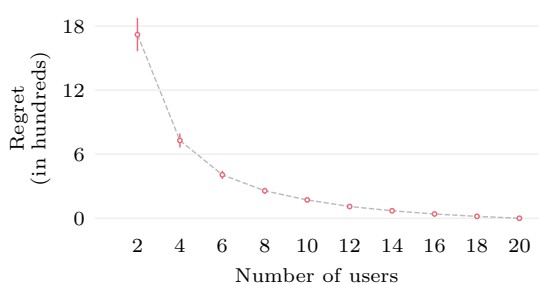

Figure 4: Expected regret incurred by `EgalUCB` over $T = 2^{18}$ time steps on Bernoulli bandits with $K = 2^{10}$ arms.

Figure 5: Expected regret incurred by `EgalUCB` over $T = 126{,}000$ time steps on Bernoulli bandits with $K = 20$ arms.

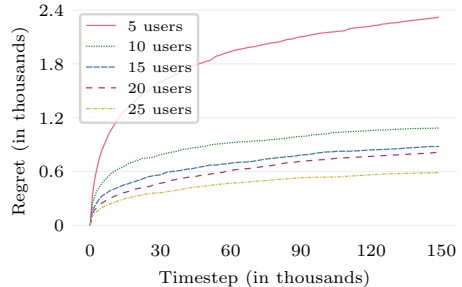

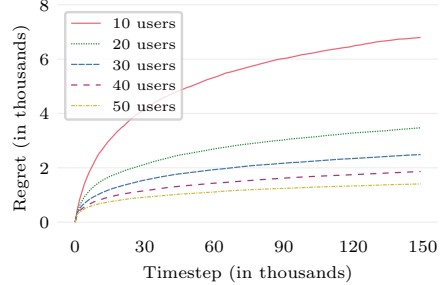

Figure 6: Expected regret incurred by `EgalUCB` over $T = 150{,}000$ time steps on the Google Cluster Usage Trace dataset with $K = 100$. Each line is associated with a different number of users $U$.

Figure 7: Expected regret incurred by `EgalUCB` over $T = 150{,}000$ time steps on the MovieLens 25M dataset with $K = 500$. Each line is associated with a different number of users $U$.

We conducted another experiment to verify the rate at which $R_T$ diminishes with $U$ when $K \gg U$. We fixed $K = 2^{10}$ and $T = 2^{18}$ while varying $U \in \{2^1, \ldots, 2^8\}$. We ran `EgalUCB` on instances of Bernoulli `EgalMAB` where $\mu_a = 0.8$ if $a \in [U]$ and $0.5$ otherwise. This choice of $\mu_a$ ensures that $\mu_*$ is kept constant for all choices of $U$. Figure 4 shows the log-log plot of $R_T$ against $U$. We observe that $R_T$ diminishes with $U$ at a rate of $O(U^{-c})$ for some constant $c \approx 1.0$. This corroborates with our policy-independent lower bound in Theorem 3 and suggests that our problem-independent upper bound in Theorem 2 may be loose.

To assess the rate at which $R_T$ diminishes with $U$ when $U \to K$, we ran a similar experiment with $T = 126{,}000$, $K = 20$, and $U \in \{2, 4, \ldots, 18, 20\}$ on Bernoulli `EgalMAB` instances. For each $U$, we ran the experiment 30 times. Figure 5 shows the plot of $R_T$ against $U$. We observe that as $U$ approaches $K$, the regret decreases to 0, and specifically when $U = K$, the regret is exactly 0. This observation aligns with the problem-independent upper bound in Theorem 2.

## 7.2 Google Cluster Usage Trace Dataset

The Google Cluster Usage Traces dataset comprises 2.4 TiB of compressed traces that record the workloads executed on Google compute cells (Wilkes, 2020). These traces are organized into tables that contain information about the machines and the instances running on them.

In our experiment, we focus on the *InstanceUsage* table from the `clusterdata_2019` trace. This table contains traces of both processor and memory usage during instance execution. To adapt to the `EgalMAB` setting, we designate each arm $a$ as a machine, uniquely identifiable using the `machine_id` field in the table. We implicitly construct its reward distribution $\mathbb{P}_a$ by drawing an entry uniformly from the trace that corresponds to the machine and return the negative of the `cycles_per_instruction` field for its reward.

Given the substantial size of the dataset, we will limit our analysis to the initial 4 million entries from the table. We pick $K = 100$ machines that contain the most amount of trace entries. Then, we formulate scenarios involving $U \in \{5, 10, 15, 20, 25\}$ *unseen* users. At each time step $t \in [T]$ where $T = 150,000$, we employ the `EgalUCB` policy to assign machines to these users.

Figure 6 shows the expected regret $R_T$ over time. Since the distribution $\mathbb{P}_a$ has bounded support, it is a subgaussian distribution. This empirical finding supports the $O(\sqrt{T \ln(T)})$ growth rate that Theorem 2 predicts.

### 7.3 MovieLens 25M Dataset

The MovieLens 25M dataset is widely used in recommender systems (Kużelewska, 2014; Forouzandeh et al., 2021) and collaborative filtering (He et al., 2017; Álvaro González et al., 2022) research. This dataset encompasses a substantial collection of 25,000,000 user ratings contributed by 162,000 users for a repository of 62,000 movies.

To adapt to the `EgalMAB` setting, we randomly select $K = 500$ movies and treat them as arms. For each movie $a \in [K]$, we implicitly construct its reward distribution $\mathbb{P}_a$ using the user ratings provided by existing users in the dataset. This empirical distribution is categorical and has support residing within $\{0.5, 1.0, \ldots, 4.5, 5.0\}$. Then, we formulate a scenario involving $U \in \{10, 20, 30, 40, 50\}$ *unseen* users. At each time step $t \in [T]$ where $T = 150,000$, we employ the `EgalUCB` policy to assign movies to these users.

Figure 7 shows the expected regret $R_T$ over time. Similar to the Google Cluster Usage Trace dataset, the distribution $\mathbb{P}_a$ is subgaussian for all machines $a$. As such, this empirical finding aligns with the $O(\sqrt{T \ln(T)})$ growth rate that Theorem 2 predicts.

## 8  Discussion

In this work, we introduced `EgalMAB`, an extension to the MAB framework with egalitarian considerations. The `EgalUCB` policy was proposed and shown to achieve an expected regret of $O\left(\sqrt{T \ln(T) \cdot (K - U) \cdot U^{-1}}\right)$. We also derived a lower bound that matches the upper bound up to a multiplicative gap of $1/\sqrt{U}$ and a term logarithmic in $T$. Our experiments on simulated and real-world data validated the theoretical analysis. Our empirical results lead us to conjecture that `EgalUCB` is indeed tight with respect to $U$. This gap could potentially be reconciled with a more refined analysis of the upper bound in future work. Other future works include:

**Adversarial semi-bandits.**   It is natural to consider the adversarial variant of our setup. After choosing a randomized assignment at each time step, an adaptive online adversary chooses the reward for each arm. Since that the set of all randomized assignments $\mathcal{B}$ (a.k.a. the Birkhoff polytope) is a convex set and the expected reward given a randomized assignment is a convex function, we conjecture that a modification of `Component Hedge` (Koolen et al., 2010) and `PermELearn` (Helmbold & Warmuth, 2009) can achieve an asymptotically near-optimal solution under the egalitarian consideration.

**Thompson Sampling.**   Another possible direction is to develop a Thompson sampling approach for the `EgalMAB` problem, potentially by adapting Combinatorial Thompson Sampling (Wang & Chen, 2018).

**Arms with capacity.**   Suppose that at each time step, we can assign at most $C$ users to each arm. When multiple users are assigned to the same arm at a given time step, we assume that they receive the same reward. This scenario particularizes to our setting when $C = 1$. Since the optimal assignment is to round-robin the top $U/C$ arms, we can redefine the regret with $\mu_* = \mu_1 + \cdots + \mu_{U/C}$. We can show that a modified version of `EgalUCB`, in which we split the horizon into $B = TC/U$ blocks and play the $U/C$ arms with the highest UCB value, achieves a regret of $R(T) \leq 2136(K - U/C)\ln(T)\Delta_{\min}^{-1} + KCU^{-1}\Delta_{\max}$ where $\Delta_{\min}$ and $\Delta_{\max}$ are redefined by replacing $U$ by $U/C$. We conjecture that this bound may be improved by dividing the horizon into $C$ phases in which we eliminate all except $K/c$ arms in phase $c \in [C]$ and play the $U/c$ arms with the highest UCB values in a round-robin fashion. If one can prove that the top $U/c$ arms survive the elimination after each phase with high probability, it is possible to achieve a factor of $C^{-1}$ in the first term in the upper bound on $R(T)$.

**Acknowledgments**

This research/project is supported by the National Research Foundation, Singapore under its AI Singapore Programme (AISG Award No: AISG2-PhD/2021-08-011). This research/project is also supported by a Ministry of Education Tier 2 grant under grant number A-9000423-00-00.

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
