# OpenReview forum: "Stochastic Bandits for Egalitarian Assignment"
_TMLR — Accepted by TMLR_

### Review · Reviewer_qkrz · 2024-08-31

**Summary Of Contributions:**

At a high level, the authors of this submission apply the multi-armed-bandit paradigm to an iterated allocation problem: given a queue of U users, a set of K>U arms (i.e. jobs and machines) and a number of rounds T, repeatedly assign users to arms in such a way that we maximize the reward of the least-rewarded user (e.g. optimize the time taken by the slowest job). After formally defining regret, the authors describe a simple extension of the UCB policy: in every block of U rounds, rotate the assignments of users to the arms with highest upper-confidence bound. The authors prove upper bounds on the regret, one with problem-dependent terms and another without them; the problem-independent upper bound is close to a lower bound they also derive. Experiments on synthetic data align with their upper bound's dependence on U, while experiments on Google Cluster Usage and MovieLens align with their upper bound's dependence on T.

**Audience:**

Yes

**Claims And Evidence:**

Yes

**Requested Changes:**

To strengthen the submission (in particular Section 6.1), consider removing the text between Lemma 3 and Lemma 5: it is quite "in the weeds" of the proof and is hard for a first-time reader to parse. Given that "The proof of Theorem 1 simply involves substituting the result of 5 into Lemma 3," it also looks like the Proof of Theorem 1 can be removed. With the space afforded by these action, you could
- add a recap of the essential notation
- provide a high-level overview of what the partition $\mathcal{G}_{b,i}$ *means* (not just the technical definition of what it *is*)
- sketch as much intuition for the proof of Lemma 5 as possible.

**Strengths And Weaknesses:**

Strengths: The use-cases provided by the authors are fairly compelling. The description of the algorithm is easy to understand as are the main theorem statements. Including the experimental results helped strengthen the contributions: there are natural applications of their algorithm and the results hint at an analysis direction.

Weaknesses: Section 6.1 is very dense with notation; I found myself having to scroll up multiple times to review the definitions. And it is hard to follow the arc of the argument because there is fairly little guiding text / intuition.

---

> ### Author Response · Authors · 2024-09-09
>
> We sincerely appreciate your time and effort in reviewing our manuscript.
>
> > Section 6.1 is very dense with notation; I found myself having to scroll up multiple times to review the definitions. You could add a recap of the essential notation.
>
> We have added a table of notation in the appendix. We hope this addition improves the readability of the section.
>
> In addition, Sections 6.1 and 6.2 are proof sketches of the problem-dependent and problem-independent bounds respectively. In fact, they can be omitted at a first reading without affecting the flow of the reader’s understanding of the main contributions of our work – i.e., the novel model that amalgamates online/sequential decision making and fairness in allocations. One can simply skip Section 6 after reading the main results in Theorems 1 to 3 and then moving on to the experimental section.
>
> > And it is hard to follow the arc of the argument because there is fairly little guiding text / intuition. You could provide a high-level overview of what the partition means (not just the technical definition of what it is) and sketch as much intuition for the proof of Lemma 5 as possible.
>
> The argument that guides the flow of the proof of the problem-dependent upper bound is provided between each lemma in Section 6, and it can be stated as follows. First, we define a "typical event" $\mathcal{E}\_{b}$ in which the empirical means are close to the true ones. This is quantified in Lemma 1. Next, we condition the analysis on the high probability good event $\mathcal{F}\_{b}$ and do a careful counting of the number of times suboptimal arms are played; this is the essence of Lemma 3. This event is further partitioned into several sub-events $\mathcal{G}\_{b,i}$. For each $i$, this is the event in which all previous $\mathcal{G}\_{b,j}$ did not happen and the number of suboptimal arms that are not played sufficiently often is greater than $\beta^{i}U$. This is a common construction used in the proof of combinatorial semi-bandits (Kveton, 2015) and serves to upper bound the high probability term in the regret by the number of times the set of arms $A\_b$ is played. This allows us to introduce the (reciprocal of the) gap term for individual arms $\Delta\_{a,N\_a}$ which then serves to derive a meaningful problem-dependent bound on the regret. We mentioned this sketch before the statement and proof of Lemma 5.

---

### Review · Reviewer_c8zg · 2024-09-02

**Summary Of Contributions:**

This work presents a novel online assignment problem. This problem a variation of the standard MAB problem, where the decision space and the objective function are different: in particular, the agent selects U actions at every round and maps them to an equal number of "users"; the goal is to maximize the cumulative reward (after T rounds) incurred by the user with the minimum cumulative reward. This kind of objective is referred to as "egalitarian", since the optimal policy would lead to U users having incurred the same cumulative reward after T rounds. The proposed algorithm is a UCB-like algorithm with an additional round-robin-like mechanism. Authors provide instance-dependent (and instance independent) regret upper bounds as well as a lower bound. UB and LB are tight w.r.t. to T but don't match for a U^{-1/2} factor. Some simulations are provided to better explain the algorithm's behavior in practice.

**Audience:**

Yes

**Broader Impact Concerns:**

I see no ethical issues.

**Claims And Evidence:**

Yes

**Requested Changes:**

- The pseudocode of the algorithm is too complex in my opinion. I understand that your pseudocode is very realistic and close to an actual code in a programming language, but for the sake of clarity is very important that your code is easily understandable. I think that you should compact some lines into a unique instruction and use a textual description of the operations instead of articulated foreach instructions. The algorithm is simple to understand so I believe that the pseudocode should also be really straightforward.
- Concerning the instance-dependent UB on regret: you state that in a standard stochastic MAB problem (U=1) you UB actually reduces to the one of UCB1. This is true, indeed, but cannot be appreciated in your bound. In the very last step of the proof, you use the inequality:
$$ \sum_{a} \frac{1}{\Delta_{a,N_a}} \leq \frac{K-U}{\Delta_{min}}.$$
In order for you to do that statement, you should also remark that for the sake of presentation you reported the bound using $\Delta_{min}$, but the actual UCB1 upper bound can be retrieved by avoiding the last upper bounding.
- Some choices of notation are a bit counterintuitive to me, see, e.g. T_{a,T}.

**Strengths And Weaknesses:**

Strengths:
- The setting is of general interest and well motivated by practical applications.
- The writing is clear and results are in general well-commented.
- The algorithm is simple to understand and implement, while the theoretical performance are also nearly-optimal.

Weaknesses:
- While the setting is interesting, I found the technical analysis very standard w.r.t. to the existing literature on stochastic MABs. The three main theoretical results follow steps that are very similar to those of standard stochastic MABs. While it is not a necessary requirement for a paper to be interesting, I found barely no technical novelty in this one.
- I would've been interested in a deeper comment about the U^{-1/2} sub optimality w.r.t. the lower bound. I agree with the authors when they conjecture that is the lower bound that is actually loose, hence the the algorithm is optimal. However, there's no hint in the paper on what could be the technical reason. I report my guess: in the proof of Lemma 8, in the very last equality, you state that:
$$ \sum_{a\in[K]} T_{a,T} = TU ,$$
I believe that this quantity should be equal to $T$, yielding the desired dependence on U^{-1/2} in the LB. Please correct me if I'm wrong here.
- The experimental campaign on real-world data provides few guidance on how data have been manipulated in order to obtain the desired data, and on how reward functions have been extracted. I would have appreciated more details on the experimental campaign in the appendix.
- While the related literature section is very rich, I would have preferred a more detailed comparison between this setting and the combinatorial bandit setting, which I believe is the closest (especially in terms of algorithms). However, the main difference between the two settings can be appreciated, so this is just a minor point.

---

> ### Author Response · Authors · 2024-09-09
>
> We sincerely appreciate your time and effort in reviewing our manuscript.
>
> > I report my guess: in the proof of Lemma 8, in the very last equality, you state that: I believe that this quantity should be equal to, yielding the desired dependence on $U^{-1/2}$ in the LB. Please correct me if I'm wrong here.
>
> The inequality in Lemma 8 is correct. Since $U$ actions are played at every timestep, the sum of $T_{a,T}$ over all actions $a$ should sum up to $TU$.
>
> > The experimental campaign on real-world data provides few guidance on how data have been manipulated in order to obtain the desired data, and on how reward functions have been extracted. I would have appreciated more details on the experimental campaign in the appendix.
>
> In Sections 7.2 and 7.3, we detailed the dataset fields used to differentiate between the arms and generate the rewards for each arm. Additionally, the full code for both experiments is provided in the supplementary materials. Should you need further clarification, we are happy to offer additional details.
>
> > I think that you should compact some lines into a unique instruction and use a textual description of the operations instead of articulated foreach instructions.
>
> We have modified the pseudocode to abstract the details of the round-robin step, which we hope improves its clarity. We also moved the original detailed pseudocode into the appendix.
>
> > You state that in a standard stochastic MAB problem (U=1) you UB actually reduces to the one of UCB1. This is true, indeed, but cannot be appreciated in your bound.
>
> We have added a note in the proof of Lemma 5 to address this technical detail and remarked this in the main text.
>
> > Some choices of notation are a bit counterintuitive to me, see, e.g. $T_{a,T}$.
>
> We recognize that some of the notation might be confusing. As such, we have added a table of notations in the appendix to make referencing the notation easier.

---

### Review · Reviewer_daK5 · 2024-09-04

**Summary Of Contributions:**

The paper introduces a new extension of the multi-armed bandit problem - EgalMAB - whereby a learner is tasked with allocating the $K$ arms to $U<K$ users such that no user is assigned the same arm. The learner's goal is to maximise the minimal expected reward for all users simultaneously. The paper introduces an algorithm (EgalUCB) based on a familiar UCB-type argument along with a cyclic permutation subroutine to account for the objective that the minimal reward across each arm is the quantity of interest. The authors prove a lower bound, and both instance dependent and worst-case regret bounds for EgalUCB, with the latter matching the lower-bound's order in $K,U$ and $T$, inclusive of log factors. Experiments are included on both simulated data and two illustrative datasets, verifying the theoretical results.

**Audience:**

Yes

**Broader Impact Concerns:**

There are unlikely to be any negative broader impacts of this work, as it is a theory paper.

**Claims And Evidence:**

Yes

**Requested Changes:**

Personally, the largest change I would like to see in the context of this work is for more substantive extensions - it seems like there were quite easy refinements which weren't made in this iteration as above mentioned. Adding at least one such extension would not be absolutely critical for my recommending acceptance, but it feels like this paper is left somewhat lacking without them.

**Strengths And Weaknesses:**

Strengths:
- The setting is natural, capturing a set of real-world settings, and the formal definition of the objective is inspired to be concordant with an established notion of fairness attributed to John Rawls, referred to as the maximin principle (Rawls himself avoids this terminology, apparently). The definition of regret captures this notion naturally using an upper bound for the maximal lowest reward over the sequence. The experiments are drawn from natural examples of settings standing to benefit from the algorithms derived, further supporting the definitions established.
- The writing is extremely clear, and results are almost certainly correct. The regret bound is extremely intuitive from the perspective of classical MAB theory, and reduced to the standard MAB setting for $U=1$.
- The introduced setting leaves room for several other natural extensions. For example, one could consider a contextual setting whereby the user's data could be considered, or a situation where there is an increased budget $B\in\{1,\dots,U\}$ on the user allocation per arm (in the current context $B=1$), which would better reflect the situation of e.g. recommending users for particular restaurants with a fixed capacity.
- Experiments are well chosen, and illustrate well the scaling of the regret with the relevant quantities, as well as demonstrate deployability of the algorithm in two practical settings.

Weaknesses:
- There does not appear to be a great deal of technical novelty in the paper, and the algorithm is a quite unsurprising extension of UCB to account for the additional structure. I did not find myself gaining any deep insights from its reading. While this may be considered a weakness, the flipside is that I have essentially no doubts about the technical correctness of the proofs, as they are founded upon established techniques.
- The paper is cantered around a relatively minimal extension of the MAB problem; I wonder why the analysis stopped here. For example, it seems like there would be minimal difficulty in including the increased budget example mentioned above. In addition, there seems to be little difficulty in an adversarial extension.

---

> ### Author Response · Authors · 2024-09-09
>
> We sincerely appreciate your time and effort in reviewing our manuscript.
>
> > There does not appear to be a great deal of technical novelty in the paper, and the algorithm is a quite unsurprising extension of UCB to account for the additional structure.
>
> We submit that it may appear that the tools used to bound the regret in both directions are, by now, standard. However, the purpose of our contribution is not to introduce new technical tools to a bandit problem; rather, it is to introduce a new bandit model that amalgamates online sequential decision making and fairness in allocating resources. The model has been acknowledged by the esteemed reviewer and the two other reviewers to be natural and motivated by practical problems. The publication of this work may inspire other researchers to develop the necessary tools to close the current gap with respect to $U$ and also to develop further extensions in terms of results and generality of the current proposed model.
>
> > The paper is centered around a relatively minimal extension of the MAB problem; I wonder why the analysis stopped here. For example, it seems like there would be minimal difficulty in including the increased budget example mentioned above. In addition, there seems to be little difficulty in an adversarial extension.
>
> We agree that the adversarial extension is an interesting one. However, as mentioned in the reply above, the purpose of the current paper is to introduce a new model and to develop some initial non-trivial results for it. Tackling the adversarial model in this paper may cloud the main message and artificially lengthen the paper. We believe that the proposed model may inspire other researchers to explore it in different directions, including adversarial models, models with corruptions, etc. We added a section to briefly discuss such extensions in the Discussion section.

---

> > ### Author Response · Authors · 2024-09-19
> >
> > We thank the reviewers for the detailed and constructive feedback. We have provided a point-to-point rebuttal as well as a careful revision of the manuscript. Should the reviewers require more clarifications, we would be happy to discuss in more detail. Thank you!

---

### Decision · Action_Editor_EMRS · 2024-09-28

**Recommendation:** Accept as is

**Comment:**

This paper studies stochastic egalitarian bandits with $K$ arms. The optimal solution to the problem is pulling the best $U$ arms in a round robin. The proposed algorithm is UCB-like. It is both analyzed and empirically evaluated.

The submitted paper was already of a high quality. Therefore, most of the comments of the reviewers were easy to address in the rebuttal. I checked some changes, such as the pseudo-code of the algorithm, and this improved in a major way. I have several additional comments based on skimming through the paper:

* Clearly state that the optimal solution is pulling the best $U$ arms in a round robin. This can be done even before you talk about bandits (end of Section 3).

* The above equivalence allows comparing your results to regret minimization with respect to the best $U$ arms. For instance, you match the gap-dependent upper bound in Theorem 2 of [Matroid Bandits: Fast Combinatorial Optimization with Learning](https://auai.org/uai2014/proceedings/individuals/161.pdf). Top $U$ arms are a uniform matroid of rank $U$. Your gap-free upper bound is better than their Theorem 3. I believe that this is because you pull only one arm per round. In general, a brief comparison of your regret bounds to those for learning the best $U$ arms would be useful.

* I agree with the reviewers that this work can be extended in many ways. One suggestion from me is looking at more general matching problems, where the mean reward of assigning a user to an arm depends on both the user and arm. This problem is also related to matroids, because it is a maximization of a modular function on the intersection of two matroids.

* How about Thompson sampling as future work?

Please take these suggestions into account in the final version.

Congratulations to the acceptance!

**Audience:**

The audience for this paper are bandit theoreticians and practitioners.

**Claims And Evidence:**

The proposed algorithm is both analyzed and empirically evaluated. The reviewers agree that the analysis is not technically challenging. This is not necessary for TMLR. On the other hand, the experiments are very well chosen to support the usefulness of the algorithm.